# Influence of Morphometry on Echocardiographic Measurements in Cavalier King Charles Spaniels: An Inverse Probability Weighting Analysis

**DOI:** 10.3390/vetsci8100205

**Published:** 2021-09-23

**Authors:** Mara Bagardi, Sara Ghilardi, Chiara Locatelli, Arianna Bionda, Michele Polli, Claudio M. Bussadori, Fabio M. Colombo, Laura Pazzagli, Paola G. Brambilla

**Affiliations:** 1Department of Veterinary Medicine, University of Milan, 26900 Lodi, Italy; sara.ghilardi@live.it (S.G.); chiara.locatelli@unimi.it (C.L.); arianna.bionda@studenti.unimi.it (A.B.); michele.polli@unimi.it (M.P.); paola.brambilla@unimi.it (P.G.B.); 2Clinica Veterinaria Gran Sasso–Milan, 20131 Milan, Italy; claudio.bussadori@gmail.com; 3Department of Veterinary Sciences for Health, Animal Production and Food Security–Lodi (Italy), University of Milan, 20122 Milan, Italy; colombofabiomaria@gmail.com; 4Centre for Pharmacoepidemiology, Department of Medicine, Karolinska Institutet-Stockholm, 17177 Solna, Sweden; laura.pazzagli@ki.se

**Keywords:** anterior mitral valve leaflet, canine morphometry, CKCS, dog, echocardiography, myxomatous mitral valve disease

## Abstract

The development and progression of myxomatous mitral valve disease (MMVD) in Cavalier King Charles Spaniels (CKCS) are difficult to predict. Thus, the identification of dogs with a morphotype associated with more severe mitral disease at a young age is desirable. The aims of this study were to: (1) describe the physical, morphometric, and echocardiographic features of class B1 MMVD-affected Cavalier King Charles Spaniels (CKCS) according to the American College of Veterinary Internal Medicine (ACVIM) guidelines; (2) evaluate the influence of morphometric physical measurements on murmur intensity, mitral valve prolapse (MVP), regurgitant jet size, and indexed mitral valve and annulus measurements. Fifty-two MMVD-affected CKCS were included in the ACVIM class B1. This is a prospective clinical cross-sectional study. Morphometric measurements, which included the body, thorax, and head sizes of each dog, were investigated to establish the association with heart murmur intensity, valvular and annular echocardiographic measurements, MVP, and regurgitant jet size, using inverse probability weighting (IPW) analyses to adjust for confounding. The IPW analyses showed that when the head length and nose length decreased, dogs had a more severe regurgitant jet size. Furthermore, subjects with a more pronounced head stop angle had thicker anterior mitral valve leaflets. A brachycephalic morphotype, as seen in dogs similar to the King Charles Spaniel breed in terms of cephalic morphology, is associated with a more severe regurgitant jet size and with valvular characteristics that are related to the most severe forms of MMVD.

## 1. Introduction

Myxomatous mitral valve disease (MMVD) is the most common acquired cardiac disease in canine patients [1]. Some studies indicate that there is a polygenically inherited component of the disease, and in at least some of the highly susceptible breeds, early predictors of MMVD development and progression, such as morphotype, could allow for improved breeding recommendations to be made [2,3,4,5,6]. As reported by Parker et al., the same genes can affect both the dogs’ body size and their heart development [7]. In fact, it is demonstrated that small-breed dogs are more predisposed to MMVD development, especially Cavalier King Charles Spaniels (CKCS) [8]. Pedersen et al. (1999) showed that there is a negative correlation between body weight and mitral valve prolapse (MVP) in this breed [9]. Furthermore, in CKCS, MMVD is associated with earlier disease onset and, thus, with potentially greater cardiac morbidity and mortality compared to other breeds [8,10,11]. However, the preclinical period often varies markedly among subjects, making it challenging for clinicians to identify those that will eventually develop clinical signs [12,13]. For these reasons, the early identification of a morphotype associated with a more severe MMVD can have several advantages. This could allow clinicians to monitor dogs in a very targeted way, as well as educating breeders regarding the selection of subjects without some phenotypical characteristics that are related to more severe MMVD and/or more rapid progression of the disease. This could be possible in the context of a breeding selection program, which should also consider all the heritable disorders of the CKCS.

To the best of the authors’ knowledge, only one study has evaluated the prevalence and severity of MVP in relation to the size of the thorax in dogs, particularly in dachshunds [3], but no study has ever analyzed the association between MVP or MMVD severity and morphometric measurements in CKCS.

The aim of this study was to investigate morphometric measurements in relation to the echocardiographic features of MMVD in ACVIM class B1 CKCS [14]. This class includes the majority of the breeding population and is very heterogeneous from the clinical, morphological, and echocardiographic points of view [15]. The identification of the phenotypic characteristics associated with more severe forms of MMVD could be useful for establishing breeding selection programs aimed at reducing the prevalence of the disease in this breed.

## 2. Materials and Methods

### 2.1. Study Population, Research Question and Statistical Framework

In this prospective clinical cross-sectional study, the authors carefully described the morphometry of a small Italian study population of CKCS and then evaluated the influence of body, thorax, and head dimensions on different clinical features (i.e., heart murmur intensity) and echocardiographic measures/indexes of the severity of MMVD (MVP, semiquantitative evaluation of regurgitant jet size, and indexed mitral valve and annulus measurements). Furthermore, they investigated the severity of MMVD through a score assigned according to the degree of MVP, mitral regurgitation jet size, and age [8]. To investigate the association between morphometric measures and the severity of MMVD, the authors used a method adopted from the causal inference framework [16]. The framework proposes methods to address causal questions while accounting for confounding, which affects the association between exposure and the outcome of interest. In this study, they used inverse probability weighting (IPW) analyses that, via weighted regression modeling, adjust for confounders [17,18]. The confounders are used to estimate the probability of being exposed, conditional on the values of the confounders; a function of this probability is used to construct weights. Weights are assigned to the subjects in the study population to balance them with respect to the confounders used in the analysis. Balancing subjects in the study population allows the estimation of an association that is unbiased from the confounders considered in the analysis while constructing the weights.

### 2.2. Inclusion Criteria and Clinical Examination

The initial population of CKCS was composed of 72 dogs. Fifty-two privately owned CKCS with asymptomatic MMVD and no cardiac enlargement (ACVIM stage B1) [14] and belonging to different lineages and breeders were recruited for enrollment in this prospective cross-sectional study. The dogs were examined during breed-health screening at the Cardiology Unit of the Veterinary Teaching Hospital, Department of Veterinary Medicine, University of Milan, between April 2019 and June 2020. Informed consent was attained from the owners, according to the ethical committee statement of the University of Milan, number 2/2016. All the dogs underwent a physical examination, echocardiography, and morphometric evaluation. The data regarding the dates of birth and the respective genealogy were verified by checking each animal’s microchip number and family tree in the Italian regional registry and the ENCI (Ente Nazionale della Cinofilia Italiana) pedigree database. Cardiac auscultation was performed by two well-trained operators (M.B. and P.G.B.) and the dogs were restrained in standing position in a quiet room by their owners. The detection of a left apical systolic murmur was not considered a mandatory inclusion criterion. The evaluated auscultatory findings were presence/absence, timing, and intensity (0 = absent; 1 = I–II/VI left apical systolic or soft murmur; 2 = III–IV/VI bilateral systolic or moderate and loud murmur, respectively; 3 = V–VI/VI bilateral systolic or palpable murmur) of murmur [19]. Unless otherwise stated, hereafter the term murmur refers to a left apical systolic murmur. The diagnosis of MMVD was based on echocardiographic evidence of changes in the mitral valve leaflets (thickening and prolapse) and the presence of mitral regurgitation on a color-flow Doppler [9]. To be included in the study, dogs must have had no evidence of left atrial and left ventricle enlargement, defined as a left atrial-to-aortic root ratio (LA/Ao) ≥ 1.6 on 2-dimensional echocardiography, and as left-ventricular normalized dimensions in diastole (LVIDad) ≥ 1.7, respectively [14]. Blood pressure was indirectly measured using the Doppler method, according to the ACVIM consensus statement [20,21].

### 2.3. Echocardiography and Assessment of Leaflet Measurements, MVP Severity and Jet Size

All echocardiograms were performed by the same operator (MB) using a MyLab50 Gold cardiovascular echocardiograph (Esaote, Genova, Italy) equipped with multi-frequency phased array probes (3.5–5 and 7.5–10 MHz), chosen according to the weight of the subject, with standardized settings. Video clips were acquired and stored using the echo-machine software for offline measurements. The exam was performed according to a standard procedure, with concurrent continuous electrocardiographic monitoring [22]. The mitral valve was evaluated using both the right and left parasternal long-axis 4-chamber views [22,23]. Valve morphology and structures, including the presence/absence and grade of valvular prolapse, were defined. The right parasternal 4-chamber view was used for the morphological evaluation of the mitral anterior leaflet during its maximum distension in the diastole [23]. The anterior mitral valve length (AMVL), width (AMVW), and area (AMVA) were measured, as well as the mitral valve annulus in the diastole (MVAd) and systole (MVAs) in the first frame after the closing and before the opening of the leaflets [23]. All the measurements were indexed according to the Wesselowski method [23]. Mitral valve prolapse was considered mild if the leaflets were prolapsing but did not cross the line joining their pivotal points (P line), moderate if they protruded between the P line and the line joining half of the echoic areas located in the lower part of the atrial septum at the level of the atrioventricular junction (T line), and severe if the leaflets exceeded the T line [24]. The sphericity index (SI) was calculated as the ratio of the LV long-axis diameter to short-axis diameter in end-diastole from the right parasternal 4-chamber long-axis view; a value of SI < 1.65 accounted for increased sphericity and was considered abnormal according to the European Society of Veterinary Cardiology (ESVC) guidelines [25,26], despite the fact that the literature has not reported SI reference values for small-breed dogs, in particular for CKCS. Left ventricular end-diastolic (EDV) and end-systolic volumes (ESV) were calculated from the right parasternal long-axis view, using the Teichholtz method, and the values were successively indexed for body surface area (BSA) in order to obtain the end-diastolic (EDVI) and end-systolic (ESVI) volume indexes [27]. The area length method was used for the calculation of 2D-derived parameters: ejection fraction (EF%), indexed for BSA 2D-EDVI and 2D-ESVI for each patient [27]. Left ventricular normalized dimensions were calculated as described by Cornell et al. (2004) [28]. The following measurements were taken from the right parasternal short-axis view: LA/Ao was obtained in the two-dimensional view as described by Hansson et al. [29], and the left ventricular diameter was measured in M-mode, with the leading-edge to the inner-edge method at the level of the papillary muscles. The color-flow mapping of the mitral valve area was obtained from the left parasternal long-axis 4-chamber view [30,31]. A pulse repetition frequency of 5 kHz was used, and the flow gain was adjusted to the maximal level without encountering background noise. The degree of MR (jet size) was assessed using color Doppler and by calculating the maximal ratio of the regurgitant jet area signal to the left atrium area (ARJ/LAA ratio) [32]. The regurgitant jet size was semi-quantitatively estimated as the percentage of the left atrial area (to the nearest 5%) that was occupied by the larger jet; it was considered to be trivial (<10%, not visible in all systolic events), trace (< 10%, present in all systolic events), mild (between 10 and 30%), moderate (between 30 and 70%), or severe (>70%) [32,33]. Echocardiographic measurements were taken by one operator (M.B.) to reduce potential biases. All measurements of interest were repeated on 3 consecutive cardiac cycles and the mean value was used in the statistical analysis [32]. The within-day intra-observer variability in the studied variables was determined by reanalyzing the parameters measured by the same observer (M.B.) 3 times after the first measurement, on a subset of 10 randomly selected blind exams from the database. The same frames from the same videos were chosen for the evaluation of intra-observer variability. The intra-observer coefficients of variation (CV range in %) were <10% for each tested variable. For descriptive purposes only, a severity score was assigned according to the degree of MVP, mitral regurgitant jet size, and age of each subject, as reported by Stern et al. in 2015, according to the formula ((Mitral valve prolapse + Regurgitant jet size) × 5)/age [34]. Due to the age-related nature of disease severity, a continuous variable was constructed so that MMVD affecting younger dogs would be considered a more severe disease variant than the one affecting older dogs, with the same level of degenerative change. The age of 5 years was chosen as a pivotal point in the breed, where dogs less than 5 years old that demonstrated clear evidence of disease were considered to be the most severely affected animals [34,35].

### 2.4. Morphometrics

Clinical and echocardiographic examinations were completed by a specific morphometric evaluation that included the assessment of the ENCI standard coat color type (Blenheim, ruby, tricolor, and black and tan) and the measurement of the body, thorax, and head of each dog. All the morphometric measurements taken, as well as their reference points, are outlined in Table 1 [36,37,38].

Body size and cephalic, thoracic, and volume indexes were also calculated (Table 1) [36]. During the morphometric evaluation, the dogs were kept calm and in a standing position by their owners, with the four limbs perpendicular, and hand stacking as if they were being exhibited. A morphometric evaluation was always performed by the same operator (M.B.) to reduce systematic errors, and on the left side of each dog to reduce potential biases. Intra-observer variability was < 10%. The circumference of the thorax was measured using a measuring tape. Body and thorax evaluations were performed using a custom-made sliding gauge (Figure 1).

Appendix A shows how the morphometric measurements were performed, with the reference points for each measurement. The dogs were classified according to the standard values reported by ENCI (https://www.enci.it/media/2405/136.pdf, accessed on 17 September 2021). Standard nose length (NL) was “about 3.8 cm”. It is important to note that the Italian breed guidelines do not give precise indications regarding the determination of the length of the nose (https://www.enci.it/media/2405/136.pdf, accessed on 17 September 2021). The body condition score (BCS) was recorded for each dog using a 1 to 9 score, and scores 4 and 5 were considered to be normal [39].

### 2.5. Exclusion Criteria

Healthy (n = 16) and MMVD-affected CKCS at ACVIM stages B2 (n = 15), C (n = 15), and D (n = 13) were not included, as well as those in stage B1 (n = 5) with either left atrial or left ventricular enlargement but not both. Dogs with cardiac diseases other than MMVD, such as myocarditis (n = 1), congenital heart defects (n = 2), cardiac tumors (n = 1), and diagnosed arrhythmias (such as supraventricular/atrial premature contractions) (n = 4), were not included in the study, as well as subjects with hypertension (n = 3) or metabolic diseases (n = 5) [21]. Dogs with facial, thoracic, and limb malformations (n = 2) and dogs younger than one year of age (n = 2) were also excluded.

### 2.6. Statistical Methods

The influence of morphometric measurements on MVP severity, jet size (ARJ/LAA ratio), murmur intensity, and echocardiographic indexed measurements (AMVL, AMVW, AMVA, MVAd, MVAs, SI) in the 52 CKCS included in the study was evaluated via IPW analyses. An IPW analysis requires the researcher to construct two regression models, one for the exposure and one for the outcome. The regression model used for the exposure is the propensity score (PS) model, which estimates, for all included subjects, the probabilities of being exposed to the risk factor of interest. For this study, the PS model included the 13 morphometric variables (exposures) as a multivariate response (“generalized multivariate propensity scores”) [18], while age, sex, weight, and coat (covariates or confounders) were used as regressors. The probabilities associated with the predicted values that resulted from the estimated PS model were the propensity score estimates. Inverse probability weights were derived as “stabilized inverse probability weights” (SIPW)—i.e., the ratio between the multivariate marginal probability density of the exposures and the multivariate probability density of the exposures conditional to covariates [40]. The second regression model is the outcome model, i.e., the main analysis model, which aims to estimate the association of the exposures with the response variables. For each type of response variable (6 numerical and 3 ordinal outcomes), a normal linear model or ordinal logistic regression models were estimated, with the morphometric variables as exposures, and the authors weighted the observations with the SIPW that had previously been estimated [41]. For the PS model, two alternative models were considered: an additive model (with no interaction between regressors) and an interaction model (with bivariate interactions between some of the regressors). For the PS model and each of the responses, the bivariate interactions between regressors were explored using normal or ordinal regression models, each of them containing one regressor at the time and only one of the possible bivariate interactions between regressors. Finally, the interaction PS model was estimated by adding all bivariate interactions that were detected as being significant in the previous exploratory models. Interaction and additive models were then compared for the best performance. The best PS model out of the additive system and the one including interactions was the model that provided a better balance of the values of the covariates across all subjects and was a better match for the positivity assumption [16]. The assumption of “positivity” implies that all participants have the potential to receive a particular level of exposure, given any value of the confounders. This was checked by identifying the range of values of the exposures, where positivity is satisfied as to the “multidimensional convex hull” (the smallest convex shape enclosing a given shape) calculated for the observed exposure values [18]. In the outcome model, robust standard errors were estimated to take the IPW into account [16]. The outcome models were checked for normality, linearity, variance homogeneity, and multicollinearity. Moreover, the outcome ordinal models were also checked for the violation of the proportional odds assumption, using surrogate residuals [42,43]. The effects of the morphometric indices that were not linear functions of some single morphometric measurements were derived from the parameters of the outcome models by means of the “delta method” [44]. All analyses were performed in an “R” environment [45].

## 3. Results

### 3.1. Clinical and Echocardiographic Results

The median age of the dogs included was 4.16 years (IQR25-75 2.91–6): 14 (27%) were younger than 3 years, 25 (48%) were between 3 and 6 years, and 13 (25%) were older than 6 years. Eleven dogs (21.2%) were intact males, 2 (3.8%) were neutered males, 34 (65.4%) were intact females, and 5 (9.6%) were neutered females. The median weight was 9.15 kg (IQR25-75 7.80–10.23). In total, 36 subjects (69.2%) weighed more than the proposed breed standard (5–8 Kg), of which 8 (22.2%) were overweight (BCS > 5) and 4 (11.1%) were underweight (BCS < 4). No subjects weighed less than the standard (5 kg). Neutered females showed a higher body weight compared to intact females and intact males (*p* < 0.05). In 26 dogs (50%), no murmurs were found. Soft murmurs (1) were present in 20 dogs (38.5%), whereas in 6 dogs (11.5%) the murmurs were of moderate/loud intensity (2). With reference to the 26 dogs with undetectable murmurs, 25 had MVP (19 mild and 6 moderate) and 19 had MR (13 trivial, 3 trace, and 3 mild). Of these 26 subjects, 18 dogs presented with MVP and MR, 1 had MR only, and 7 had MVP only. However, 51 (98.1%) of the 52 subjects included had MVP. The sphericity index was lower than 1.65 in 48 (92.3%) subjects. Table 2 shows all clinical data (age, body weight and sex); indexed mitral valve measurements; and MVP, jet size, murmur severity, and severity scores for all the subjects included. Moreover, MVAd was greater in subjects over 6 years old than in dogs younger than 3 years (*p* = 0.03), whereas MVA was greater in subjects over 6 years old than in those aged between 3 and 6 years (*p* < 0.001). The sphericity index was lower in subjects over 6 years than it was in subjects aged between 3 and 6 years (*p* = 0.01).

### 3.2. Morphometric Measurements

Forty-four (84.6%) dogs, 11 (21.1%) males and 33 (63.5%) females, had a height at the withers that was lower than the breed standard (34–36 cm for males and 32–35 cm for females). In 6 (11.5%) subjects, the nose length was longer than the standard (3.8 cm), whereas in 45 (86.5%) it was shorter than 3.8 cm. In 36 CKCS (69.2%), the nose length was shorter than 3.5 cm and in 17 (32.7%) it was shorter than 3 cm. In only one subject was the nose length equal to the standard measure (3.8 cm). The morphometric measurements, coat color type, and BCS of the overall included population are shown in Table 2. The morphometric indexes are reported in Table 2. Neutered females showed a greater thorax height, thorax width, TC3, and volume index than intact females, and a greater TC3 and volume index compared with intact males (*p* < 0.05). Furthermore, intact males showed a greater height at the withers, head length, and head–nose length compared with intact females (*p* < 0.05). No differences between nose length, head width, and head stop angle were found between the sexes (*p* > 0.05). The head stop angle was lower (i.e., closer to 90°) in subjects with a weight within the standard range (5–8 Kg) (*p* = 0.04). Subjects with the tricolor coat type had a greater head width than the ones with a Blenheim coat type (*p* = 0.01). Furthermore, younger subjects (<3 years) weighed less than older subjects (*p* < 0.001), showed a lower thorax length (*p* < 0.05) and volume index (*p* < 0.01), and showed a higher severity score (*p* < 0.05).

### 3.3. Settings for IPW Analysis

The IPW analysis was performed, including only 49 of the 52 subjects, due to the lack of some values for ordinal and continuous variables. Furthermore, the covariate “coat color type” was simplified in the categories “Blenheim” and “other colors” due to the small number of subjects with coats that were not Blenheim. For the same reason, the information “neutered/sterilized”, which was originally incorporated in the covariate “sex”, was not used. The PS model with interactions between covariates/confounders provided a better balance in terms of the confounders observed between exposure groups (increasing comparability); accordingly, this model was used to build SIPW.

### 3.4. IPW Analyses for Ordinal Variables

The results obtained were significant for two of the three considered ordinal variables, particularly for heart murmur intensity and jet size. The IPW analysis, in fact, showed that body length (*p* = 0.03) and nose length (*p* < 0.01) had negative influences on heart murmur intensity (a shorter body length and shorter nose were associated with a higher murmur intensity). Furthermore, head length (*p* < 0.001) had a negative influence on jet size (shorter head was associated to larger jet size). However, morphometric measurements had no effects on MVP severity. The results of the regression analysis for ordinal variables, as applied to the included population, are summarized in Table 3.

### 3.5. IPW Analyses for Continuous Variables

Head length (*p* < 0.001) had a positive influence on the anterior mitral valve length (a longer head was associated with a longer anterior mitral valve leaflet). The thorax width (*p* = 0.01) had a positive influence on the anterior mitral valve width, whereas thorax length (*p* = 0.04) had a negative one: subjects with a larger or shorter thorax had a thicker anterior mitral valve leaflet. The variables of body length (*p* = 0.02), thorax width (*p* = 0.000), mean or papillary circumference (TC2) (*p* < 0.001), head length (*p* = 0.000), and head stop angle (*p* = 0.000) had positive influences on mitral valve annulus in the diastole, whereas thorax height (*p* = 0.02), TC1 (*p* = 0.000), TC3 (*p* = 0.02), and nose length (*p* < 0.001) had a negative influence. Thorax width (*p* = 0.01) and head stop angle (*p* = 0.000) had positive influences on the mitral valve annulus in the systole, whereas thorax height (*p* = 0.002) had a negative influence. Anterior or axillary thoracic circumference (*p* = 0.01) and head length (*p* = 0.000) had a positive influence on the sphericity index. The derivation of the results of the IPW analysis with respect to the morphometric indexes showed that only the thoracic index was negatively associated with a mitral valve annulus in the systole and diastole. The results of the regression analysis for continuous variables, as applied to the included population, are summarized in Table 3. The performance of the outcome models regarding normality, linearity, and variance homogeneity were acceptable according to the diagnostic plots, and there was no indication of the violation of the proportional odds assumption for the ordinal models, whereas some multicollinearity problems were detected in the models, including interaction terms (data not shown).

## 4. Discussion

Very little is known regarding the relationship between echocardiographic indicators of the severity of MMVD (i.e., MVP severity, jet size, and indexed echocardiographic measurements) and the morphometric measures for all breeds. To the best of the authors’ knowledge, this study is the first-ever on the relationship between morphometric data and echocardiocolor Doppler measures in CKCS to have been carried out. The highlighting of any association between morphometric data, the severity of echocardiographic lesions, clinical symptoms, and the evolution time of the disease will be reached only through a long-term follow-up of the subjects and a longitudinal study. Through their results, the authors have tried to lay the basis for this. In the present study, is clear that subjects with a larger thorax width or a shorter thorax length (more barrel-shaped) and a shorter head had thicker anterior mitral valve leaflets. The mitral valve annulus in the diastole has been observed to be larger in subjects with a smaller thorax height (reduced dorsoventral thoracic dimension), larger thorax width, and greater mean or papillary thoracic circumference (TC2). The same was observed in subjects with shorter noses. Regarding the mitral valve annulus in the systole, the results are superimposable to the diastole: the annulus has been observed to be greater in subjects with a smaller thorax height and larger thorax width. It is also interesting to underline the positive influence of anterior or ancillary thoracic circumference (TC1) and head length on the sphericity index: the ventricular shape was more spherical in subjects with a smaller TC1 and shorter head. These findings are similar to those observed for human medicine, in which MVP is associated with an asthenic habitus, corresponding to a reduced antero-posterior thorax diameter [46,47]. In fact, asthenic habitus in dogs could be considered as a reduction in the dorsoventral diameter, given the quadrupedal station. Furthermore, in the study carried out by Olsen in dachshunds [3], the thoracic circumference was found to be negatively correlated with the severity of MVP. Obviously, from the results obtained, the authors can only speculate about the influence of some morphometric measures on heart murmur intensity and jet size, but not the influence on MVP. However, as observed by Olsen [3], the authors may suppose that the results obtained may indicate an echocardiographic phenotype that is more easily associated with mitral valve disease (shorter or thicker anterior mitral valve leaflets, greater mitral valve annulus in the systole and diastole, and lower sphericity index). Only one study in CKCS had shown a negative correlation between the severity of MVP and body weight, demonstrating that smaller dogs have more severe forms of MMVD [9]. With these results, as stated before, the authors are not able to demonstrate the same; on the other hand, the association between the cranial morphology of the subjects, the severity of the heart murmur, the jet size dimension, and other valvular characteristics is still relevant. The authors observed that subjects with a shorter head are more likely to have a higher jet size. Furthermore, subjects with a shorter body and nose length have a greater heart-murmur intensity. Thus, given all the results discussed, CKCS that have a shorter nose and head, and a more barrel-shaped thorax, are likely to have valvular characteristics related to more severe forms of MMVD than are those dogs with longer and narrower skulls and bodies. According to these results, the breeding of subjects with a cranial morphology tending toward brachycephalism (wider and shorter head) may be counterproductive in view of the selected reproduction for MMVD, although additional studies are needed to confirm the authors’ findings. In the literature, only one study has investigated the influence of coat type (length) on MVP prevalence and severity, particularly in dachshunds [3]. No association between MMVD and coat color has ever been described. In the present study, coat color, taken as a single factor, did not affect any of the MMVD indexes. Nevertheless, it should be considered that, in the sample included in this study, 68% of the subjects were Blenheim: in Italy, this is by far the most common color (65%), as reported by ENCI (https://www.enci.it/libro-genealogico/razze/cavalier-king-charles-spaniel, accessed on 17 September 2021). The association between MMVD and coat color type should be evaluated in a larger population of dogs, including subjects in more advanced ACVIM classes. Many of the dogs included had mild changes whose long-term significance is unknown, due to the lack of data from a follow-up. In this study, as already stated, a high percentage of dogs without murmurs had mild to moderate MVP, and the authors report a high prevalence of echocardiographically detected MR in CKCS with no murmurs, reinforcing the notion that a purely clinical screening is unable to identify MMVD in this breed [48]. Similarly, subjects with slight mitral valve degeneration may have higher-intensity murmurs, indicating that there is no basis for proposing that murmur intensity is a valid indicator of disease severity. These findings highlight the need for additional research to follow up these test subjects and establish which of them will develop the disease with a faster course and which ones will remain stable with mild forms until an advanced age. Furthermore, this will enable us to understand whether some physical characteristics can be related to the progression of MMVD and, bearing in mind the results of this study, increase the focus on the evaluation of thorax and skull dimensions of the included subjects in follow-up research. In the end, the morphometric data obtained, combined with the genetic analysis and echocardiographic evaluation of the subjects, could help to characterize some phenotypes related to more severe forms of MMVD. Among the strengths of this study, the authors note that the accurate statistical approach used, based on IPW analyses, allowed them to account for the measured confounding variables in the association between morphometric measures and MMVD. However, this study also had some limitations. The qualitative nature of some echocardiographic parameters means that they should be interpreted with caution. Jet size, as detected by color flow mapping, should only be regarded as a semiquantitative measure of the degree of MR, unlike methods using the proximal isovelocity surface area (PISA) and the vena contracta [49]. Several factors, such as the quality of the imaging window, the distance to the flow being imaged, gain settings, pulse repetition frequency setting for the color Doppler test, the immobility of the patient, and the experience of the operator, may influence this measure [50]. The left apical 4-chamber view was used because the degree of MR may be underestimated if color flow mapping is performed from the right side of the thorax [49]. Furthermore, it has long been known that, given the 3D morphology of the mitral valve, long-axis images that do not include the left ventricular outflow tract (LVOT) greatly overestimate the presence of MVP in humans. Considering the recent publications on canine 3D mitral valve morphology [51] in both multi-breed populations and CKCS, this cannot be ignored any longer. It must be highlighted that many of these dogs may not really have mitral valve prolapse. Therefore, specific studies assessing the association between quantitative echocardiographic severity measurements of MMVD and the morphometry of subjects are necessary. To the best of the authors’ knowledge, SI cut-off reference values for small-breed dogs have not been reported in the literature. Due to the limits of the cut-off value chosen, the obtained results suggest that CKCS have a ventricular morphology that is extremely different from all breeds described in the guidelines; thus, specific SI values are needed for CKCS [25,26]. Eventually, further investigations using a greater number of subjects with a non-Blenheim coat color type and in advanced ACVIM stages are needed, as well as with a higher number of males and subjects that are not overweight. It is also necessary to point out that the subjects included were of different ages at the time of diagnosis. Via IPW analyses, this study attempted to find an association between morphometric measures and MMVD that was unbiased by the confounding factors of age, sex, weight, and coat type, but the presence of unmeasured relevant confounding is still possible.

## 5. Conclusions

In the CKCS included in the present study, MVP had an epidemiology resembling that known for MVP in humans and dachshunds [3,46,47]. In fact, MVP severity was significantly positively associated with measures of the degree of MMVD (e.g., jet size, leaflet length, and murmur intensity). In the present study, thorax height had a negative association with AMVL. Furthermore, thorax width and TC1 had positive associations with MVAd and SI, respectively. In particular, the most interesting result obtained is that subjects with a shorter head were associated with a higher jet size, while subjects with a shorter body and nose length had a greater heart murmur intensity. Regarding mitral valve and mitral annulus measurements, subjects with a more barrel-shaped thorax and a shorter nose had shorter and thicker anterior mitral valve leaflets and a greater mitral valve annulus in the systole and diastole. This suggests that a brachycephalic morphotype, with dogs that are much more similar to the King Charles Spaniel breed regarding cephalic morphology, is correlated with a more severe jet size and with valvular characteristics related to more severe forms of MMVD; this may be counterproductive in view of the selected reproduction for MMVD. Studies focusing on the follow-up of B1 subjects, and on the results of this study, would allow researchers to gain a better understanding of the morphological aspects that are often associated with the more serious and/or faster evolution of the disease in the CKCS. It could be useful to collect additional information relating to more advanced ACVIM classes and older subjects. This, together with clinical and echocardiographic characterization, could be used as part of a screening program for CKCS, defining early selection criteria for the exclusion of a subject from reproduction.

## Figures and Tables

**Figure 1 vetsci-08-00205-f001:**
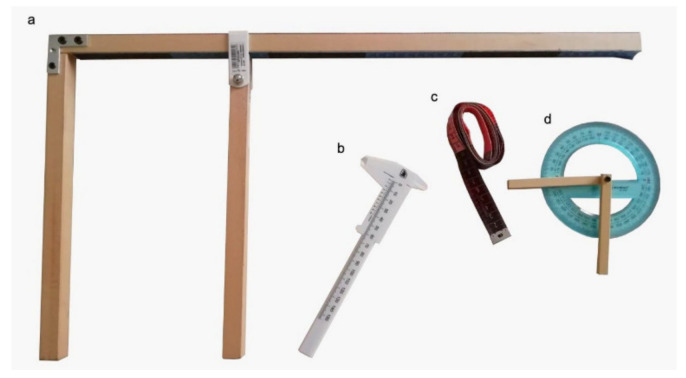
Measurements tools used to evaluate the thorax and body dimensions. (**a**) The basal part of the custom-made sliding gauge used to measure the height at the withers and the width and height of the thorax and the mobile part that, together with the basal part, was used to measure the height at the withers and width and height of the thorax. (**b**) The gauge used to measure the head length, nose, and head width. (**c**) The measuring tape used to measure the three thorax circumferences. (**d**) The goniometer used to measure the head’s stop angle.

**Table 1 vetsci-08-00205-t001:** Morphometric measurements, body indexes, and reference points [36,37,38].

**Body Measurements**	**Thorax Measurements**	**Head Measurements**
Height at the withers (WH): distance of the withers from the ground, measured at the top of the shoulder blades.	Height (TH): distance between the back and the sternum, measured behind the shoulders.	Head length (HL): measured from the top of the occipital ridge to the horizontal line joining the two inner corners of the eyelids.
Body length (BL): distance between the tip of the shoulder and the tip of the buttock.	Width (TW): measured just behind the shoulders.	Nose length (NL): measured from the horizontal line joining the two inner corners of the eyelids to the cranial extremity of the truffle.
Width at the chest (CW): measured at the shoulder–humeral joints.	Length (TL): distance between the shoulder tip and the midline of the last rib.	Head + nose length (HNL): HL + NL.
	Circumference (TC_1_, TC_2_, TC_3_): - TC_1_ (anterior or axillary circumference): measured at the level of the anterior part of the axillary cable. - TC_2_ (mean or papillary circumference): measured at the level of the first mammary nipples. - TC_3_ (lower or basal circumference): measured at the level of the xiphoid process of the sternum, which corresponds above the spinous process of the seventh cervical vertebra.	Head width (HW): measured at the zygomatic arches.
		Head stop angle (HA): angle obtained, with the head seen in profile, by the intersection of a line tangent to the frontal region (between the two orbits) and the line of the upper part of the nasal barrel.
Body indexes		
Cephalic index: (HW × 100)/HNL.		
Craniofacial ratio (CFR): NL/HL.		
Thoracic index: (TW × 100)/TH; height thorax index: (TH × 100)/WH.		
Volume index: (body weight × 100)/WH.		
Body size: (WH × 100)/BL.		

**Table 2 vetsci-08-00205-t002:** Clinical data showing indexed mitral valve measurements, MVP, jet size, murmur severity, severity score, morphometric measurements, indexes, coat color type, and BCS of all the subjects included.

(a) Clinical data, indexed mitral valve measurements, MVP, jet size, murmur severity, and score of severity												
Age (y)	Body weight (Kg)	Sex	AMVL (cm)	AMVW (cm)	AMVA (cm)	MVAd (cm)	MVAs (cm)	SI	MVP	Jet size	Heart murmur severity	Score of severity
4.16 (2.91–6.00)	9.15 *^,†^ (7.80–10.23)	F (n.35) NF (n.5) M (n.11) NM (n.2)	0.70 (0.63–0.79)	0.14 (0.12–0.16)	0.08 (0.06–0.11)	0.78 (0.74–0.88)	0.61 (0.55–0.66)	1.37 (1.24–1.50)	0 (n.1) 1 (n.31)2 (n.19)3 (n.1)	0 (n.6) 1 (n.18) 2 (n.7) 3 (n.14)4 (n.7)	0 (n.26) 1 (n.20) 2 (n.6)	2.69 (2.01–3.45)
(b) Body and thoracic morphometric measurements												
Body morphometric measurements						Thoracic morphometric measurements						
WH		BL		CW		TH		TW	TL	TC_1_	TC_2_	TC_3_
29.20 (27.78–31.58)		33.75 (29.70–35.85)		12.25 (11.45–13.53)		14.95 (13.58–16.13)		11.95 (11.28–13.00)	20.15 (18.68–22.10)	47.00 (45.00–49.63)	47.50 (46.00–50.63)	45.50 (43.58–49.50)
(c) Head morphometric measurements and physical data												
Head morphometric measurements											Physical data	
HL		NL		HNL		HW			HA		Coat color type	BCS
7.70 (7.20–8.13)		3.20 (2.80–3.50)		10.95 (10.20–11.85)		7.75 (7.50–8.10)			115.00 (110.00–120.00)		(n. 35) B (n. 4) B&T (n. 2) R (n. 11) T	(n. 6) 3 (n. 10) 4 (n. 26) 5 (n. 10) 6
(d) Body indexes												
Cephalic index			Craniofacial ratio (CFR)		Thoracic index		Height thorax index		Volume index		Body size	
71.97 (68.08–75.50)			0.40 (0.36–0.44)		83.08 (75.54–86.83)		0.5 (0.49–0.53)		30.82 (25.97–34.31)		88.40 (83.54–93.09)	

(a) Note: The severity score formula was calculated as ((Mitral valve prolapse + Regurgitant jet size) × 5)/Age [35]. All echocardiographic measurements were indexed to body weight using the scaling exponents calculated by Wesselowski, one for each specific valve measure [24]. They were, respectively, 0.37, 0.41, 0.78, 0.37, and 0.40 for AMVL, AMVW, AMVA, MVAd, and MVAs [21]. The variables are reported as median and interquartile ranges (IQR: 25th to 75th). Abbreviations: F = intact females; NF = neutered females; M = intact males; NM = neutered males; AMVL = anterior mitral valve length; AMVW = anterior mitral valve width; AMVA = anterior mitral valve area; MVAd = mitral valve annulus in diastole; MVAs = mitral valve annulus in systole; SI = sphericity index; MVP = mitral valve prolapse severity (0 = absent, 1 = under P line, 2 = between P and T lines, 3 = over T line); jet size = mitral regurgitation severity (0 = absent; 1 = trivial, regurgitant jet area (ARJ)/left atrial area (LAA) < 10% and not present in all cardiac cycles; 2 = trace, ARJ/LAA < 10% and present in all cardiac cycles; 3 = mild, 10% < ARJ/LAA < 30%; 4 = moderate, 30% < ARJ/LAA < 70%); heart murmur severity: 0 = absent; 1 = I-II left systolic or soft murmur; 2 = III-IV bilateral systolic or moderate and loud murmur, respectively; 3 = V-VI bilateral systolic or palpable murmur. * Values were significantly higher (*p* < 0.05) in NF compared to F; ^†^ values were significantly higher (*p* < 0.05) in NF compared to M. (b) Note: The variables are reported as medians and interquartile ranges (IQR: 25th to 75th). All morphometric measurements are reported in cm. Abbreviations: WH = height at withers; BL = body length; CW = width at chest; TH = thorax height; TW = thorax width; TL = thorax length; TC1 = thoracic anterior or axillary circumference; TC2 = thoracic mean or papillary circumference; TC3 = thoracic lower or basal circumference. (c) Note: The variables are reported as medians and interquartile ranges (IQR-25th to 75th). All morphometric measurements are reported in cm, whereas HA is reported as °. Abbreviations: HL = head length; NL = nose length; HNL = HL + NL; HW = head width; HA = head stop angle; B = Blenheim; B&T = black and tan; R = ruby; T = tricolor; BCS = body condition score (scores of 4 and 5 were considered to be normal). (d) Note: the variables are reported as medians and interquartile ranges (IQR: 25th to 75th).

**Table 3 vetsci-08-00205-t003:** Results of the regression analysis applied to determine the influence of morphometric variables on the clinical and echocardiographic parameters of all included subjects.

Regression Analysis										
	BL	TH	TW	TL	TC_1_	TC_2_	TC_3_	HL	NL	HA
a. Ordinal variables										
MVP										
Jet size								** n		
Heart murmur intensity	* n								** n	
b. Continuous variables										
AMVL								* p		
AMVW			* p	* n						
AMVA										
MVAd	* p	* n	*** p		*** n	*** p	* n	*** p	*** n	*** p
MVAs		** n	* p							*** p
SI					** p			** p		

Note: All morphometric measurements are expressed in cm, whereas HA is expressed as °. Morphometric variables not influencing the clinical and echocardiographic parameters are not reported in the table. Level of statistical significance: *** = *p* < 0.001; ** = *p* < 0.01; * = *p* < 0.05. p = positive association; n = negative association. Abbreviations: MVP = mitral valve prolapse severity; jet size = severity of mitral regurgitation; anterior mitral valve: length (AMVL), width (AMVW), area (AMVA); MVAd = mitral valve annulus in diastole; MVAs = mitral valve annulus in systole; SI = sphericity index; BL = body length; TH = thorax height; TW = thorax width; TL = thorax length; TC1 = thoracic anterior or axillary circumference; TC2 = mean or papillary circumference; TC3 = thoracic lower or basal circumference; HL = head length; NL = nose length; HA = head stop angle.

## Data Availability

The data presented in this study are all reported in the manuscript.

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
