# Peer review of "Influence of Morphometry on Echocardiographic Measurements in Cavalier King Charles Spaniels: An Inverse Probability Weighting Analysis"

_vetsci, 2021, doi:10.3390/vetsci8100205_

Round 1
Reviewer 1 Report
This a very well written cross sectional study about the correlation between morphological physical features and clinical expression of chronic Mitral Valve Disease in Cavalier King Charles Spaniel (CKCS) dogs in stage B1. A smart panel of morphometric parameters have been accurately described as well as the methodology about how to measure them. This is a very novel approach to cardiac disease in veterinary medicine that could be very useful for veterinary practitioners and for CKCS dogs breeders. I am an enthusiastic about this approach that gives to the physical examination its own dignity and importance and that I would call “cardiology beyond echocardiography”. The study is thoroughly well designed and discussed with the accompanying images nicely informative. The weaker point of the study is the population of dogs enrolled, in which intact females and overweight subjects are overrepresented with the risk of a potential sample bias. This can be particularly important for a study in which morphometric parameters are under investigation. The other limit, well discussed by the authors, could be addressed by proceeding with a longitudinal study about this topic, as the meaning of early mild echocardiographic changes that leads to “only echocardiographically detectable” mitral regurgitation in CKCS is still unclear.
Please consider the following suggestion:
Line 108: substitute “dilatation” with “enlargement”
Line 114: please correct “operators” with “operator” and indicate the initials for consistency.
Line 139: erase “from the short axis”
Line 142-146 please indicate in which scan view you obtained the measurements. If it is from the right parasternal long axis 4 chambers view consider moving this part at line 137.
Author Response
This a very well written cross sectional study about the correlation between morphological physical features and clinical expression of chronic Mitral Valve Disease in Cavalier King Charles Spaniel (CKCS) dogs in stage B1. A smart panel of morphometric parameters have been accurately described as well as the methodology about how to measure them. This is a very novel approach to cardiac disease in veterinary medicine that could be very useful for veterinary practitioners and for CKCS dogs breeders. I am an enthusiastic about this approach that gives to the physical examination its own dignity and importance and that I would call “cardiology beyond echocardiography”. The study is thoroughly well designed and discussed with the accompanying images nicely informative. The weaker point of the study is the population of dogs enrolled, in which intact females and overweight subjects are overrepresented with the risk of a potential sample bias. This can be particularly important for a study in which morphometric parameters are under investigation. The other limit, well discussed by the authors, could be addressed by proceeding with a longitudinal study about this topic, as the meaning of early mild echocardiographic changes that leads to “only echocardiographically detectable” mitral regurgitation in CKCS is still unclear.
The authors thank the Reviewer for the thoroughly review of our study. We are very pleased about the appreciation of the work and the positive comments. We are aware of the limits mentioned by the Reviewer, and we have better specified in the discussion the limit related to the weight and the sex of the enrolled subjects Moreover, the analyses have been performed adjusting for the confounding factors “age, sex, weight, and coat” which were used as regressors in the propensity score model. The propensity score model has been furtherly used to weight subjects in the study population on the base of such confounding factors making them more comparable and avoiding the confounding bias deriving by differences in age, sex, weight and coat of the dogs included in the analysis. With this method the differences in weight and sex should not constitute any longer a source of bias.
Please consider the following suggestion:
Line 108: substitute “dilatation” with “enlargement”
Thank you for the suggestion, we have substituted this word.
Line 114: please correct “operators” with “operator” and indicate the initials for consistency.
Thank you, we have corrected and added the initials of the operator.
Line 139: erase “from the short axis”
Thank you, done.
Line 142-146 please indicate in which scan view you obtained the measurements. If it is from the right parasternal long axis 4 chambers view consider moving this part at line 137.
Thank you, we have added the information regarding the scan view and moved this part at line 137. We have also changed the numeration of the bibliographic references.
We feel that the quality of our manuscript has improved following the Reviewers’ comments and suggestions Thank you very much.
Best regards

Reviewer 2 Report
The authors present an intriguing study examining the association of various physiognomic characteristics of CKCS and various derangements of the mitral valve in dogs with mild mitral valve disease. The premise of the study is that certain morphological traits might be reflected in either earlier onset of mitral valve disease, or more rapid progression.
I am not qualified to determine the veracity of the statistical approach, so my review will be limited to the clinical aspects.
The authors identified several mitral valve characteristics that appeared to associate with surface characteristics. Specifically body and nose length were associated with murmur intensity and head length was associated with jet size.
I have a few concerns about the variables the authors chose to examine. Jet size was determined as ARJ/LAA. However, this method does not evaluate the regurgitant volume – jet area is a function of turbulence and velocity of flow, and has been shown to correlate poorly with regurgitant volume. Indeed, given that all the dogs in this study had normal LA and LV, the regurgitant volumes in all dogs would have to be very small. The authors’ own data underscore the invalidity of using ARJ/LAA as an estimate of regurgitant volume or MR “severity”. This concern also invalidates the severity index proposed by Stern et al.
Similarly, murmur intensity might not correlate with disease severity as it depends on several complex physical factors, the most important of which is the jet angle. Jets that are directed eccentrically along the lateral border of the left atrium will tend to be louder than those directed medially (along the septum) or centrally. Again, the authors’ data support the observation made by other investigators that dogs with very mild disease can have moderate or loud murmurs. Therefore, there is no basis for proposing that murmur intensity is a valid measure of disease severity.
Therefore, the significant findings likely do not reflect disease severity. If the authors remove those variables from the analysis, I suspect they will be left without any differences. Furthermore, if MVP severity correlated with jet size and murmur intensity, why did MVP not correlate with any morphometric variables?
My other concern is that the authors did not measure these morphometric/physiognomic variables in older CKCS that had no evidence of mitral valve disease (i.e. healthy dogs). Finding that the specific variables (head length, nose length and body length) might segregate with B1 dogs that had more “mild” disease would support the authors’ arguments.
The authors describe the study as cross-sectional – this is reasonable, but causal associations cannot be easily made with such a study design.
An alternative approach would be to take a population of CKCS with varying degrees of disease severity (from mild to severe) based on traditional measures of severity – LA volume, LV volume etc – and compare the morphometric characteristics in age matched and sex-matched dogs. In other words, do four 5-year old male dogs that have zero, mild, moderate and severe disease differ? With a large enough population that is age-and-sex-matched, the authors might find some traits that change with disease severity.
Finally, to show that any traits are associated with severity or “malignancy” the authors would need to perform a longitudinal study demonstrating that those dogs with shorter bodies, shorter nose and shorter heads developed CHF sooner than age-matched counterparts with longer noses, bodies and heads.
Minor comments
Table 2 – many of the data are missing, at least based on the Table title. No mitral valve values were provided to this reviewer. I could only see morphometric data.
Author Response
The authors present an intriguing study examining the association of various physiognomic characteristics of CKCS and various derangements of the mitral valve in dogs with mild mitral valve disease. The premise of the study is that certain morphological traits might be reflected in either earlier onset of mitral valve disease, or more rapid progression.
I am not qualified to determine the veracity of the statistical approach, so my review will be limited to the clinical aspects.
The authors thank the Reviewer for the thoroughly review of our study. Thank you very much.
We have carefully considered all Reviewer’ comments and have tried to address them whenever we felt this was appropriate.
The authors identified several mitral valve characteristics that appeared to associate with surface characteristics. Specifically body and nose length were associated with murmur intensity and head length was associated with jet size.
I have a few concerns about the variables the authors chose to examine. Jet size was determined as ARJ/LAA. However, this method does not evaluate the regurgitant volume – jet area is a function of turbulence and velocity of flow, and has been shown to correlate poorly with regurgitant volume. Indeed, given that all the dogs in this study had normal LA and LV, the regurgitant volumes in all dogs would have to be very small. The authors’ own data underscore the invalidity of using ARJ/LAA as an estimate of regurgitant volume or MR “severity”. This concern also invalidates the severity index proposed by Stern et al.
Thank you very much for this comment, we are aware of the limit of the ARJ/LAA compared to PISA or vena contracta methods. We have added this information in the discussion as a limit, when we declared that jet size detected by color flow mapping should only be regarded as a semiquantitative measure of the degree of MR. However, given the complete absence of other studies related to this evaluation (except for the work of Dr. Olsen in the Dachshunds in 1999) we thought of using Stern’s formula, and consequently the ARJ/LAA method, as a basis to build our study. Because of the age-related nature of disease severity, we needed to get a continuous variable that would allow us to consider a MMVD affecting younger dogs as a more severe disease variant than the one affecting older dog with the same level of degenerative change.
Similarly, murmur intensity might not correlate with disease severity as it depends on several complex physical factors, the most important of which is the jet angle. Jets that are directed eccentrically along the lateral border of the left atrium will tend to be louder than those directed medially (along the septum) or centrally. Again, the authors’ data support the observation made by other investigators that dogs with very mild disease can have moderate or loud murmurs. Therefore, there is no basis for proposing that murmur intensity is a valid measure of disease severity.
Thank you very much for this specification. We have changed the introductive sentence of the material and method section to be clear that the intensity of the heart murmur is not an indicator of the severity of the disease, the phrase could mislead and now has changed: “In this prospective clinical cross-sectional study, we carefully described the morphometry of a small Italian study population of CKCS and then we evaluated the influence of body, thorax, and head dimension on different clinical features (i.e., heart murmur intensity) and echocardiographic measures/indexes of the severity of MMVD (MVP, semiquantitative evaluation of regurgitant jet size, and indexed mitral valve and annulus measurements).”
The same thing was better explained in the discussion in two different parts (n. 2 relating also to the previous Reviewer’s comment):
- “Many of the included dogs had mild changes, whose long-term significance is un-known due to the lack of data about follow-up. In this study, as already published, a high percentage of dogs without murmurs had MVP from mild to moderate and we report a high prevalence of echocardiographically detected MR in CKCS with no murmurs, reinforcing the inability of a purely clinical screening to identify the MMVD in this breed [48]. Similarly, subjects with slight mitral valve degenerations may have higher intensity murmurs, supporting that there is no basis for proposing that murmur intensity is a valid indicator of disease severity. These proportions evoke the need of additional research to follow-up these subjects and understand which of them will develop the disease with a faster course and which ones will remain stable with mild forms up to advanced age.”
- “Considering recent publications on canine 3D mitral valve morphology [51], both in multi-breed populations and specifically in CKCS, this cannot be ignored any longer. It must be highlighted that many of these dogs may not really have mitral valve prolapse. Therefore, specific studies assessing the association between quantitative echocardiographic severity measurements of MMVD and morphometry of subjects are necessary.”
Therefore, the significant findings likely do not reflect disease severity. If the authors remove those variables from the analysis, I suspect they will be left without any differences. Furthermore, if MVP severity correlated with jet size and murmur intensity, why did MVP not correlate with any morphometric variables?
Thank you for this comment. We have tried to resolve the expressed doubts. As previously specified, the clinical and echocardiographic parameters evaluated in this study were chosen based on what was already published in the veterinary literature; the discussion and some sentences previously included in this cover letter were modified based on the comments of the Reviewer. Thank you. In addition, only a study in dachshunds (Olsen, 1999) and CKCS (Pedersen, 1999) managed to find a significant correlation between respectively chest size and MVP and weight of subjects and MVP, supporting the assumptions reported for humans. This should certainly be more investigated on a larger population of subjects, especially with their follow-up. In our opinion, this work is the basis for the construction of future studies more focused on the echocardiographic quantitative evaluation of MMVD.
My other concern is that the authors did not measure these morphometric/physiognomic variables in older CKCS that had no evidence of mitral valve disease (i.e. healthy dogs). Finding that the specific variables (head length, nose length and body length) might segregate with B1 dogs that had more “mild” disease would support the authors’ arguments.
Thank you for this comment. We have a clinical population of subjects, and we have no healthy subjects older than 4 years of age. Furthermore, healthy subjects are a small number, not statistically significant for a comparison with affected dogs. This is the reason why we have stressed about the follow up in the conclusion section, changing this sentence: “Studying of the B1 subjects’ follow-up and the results of this study would allow for a better understanding of the morphological aspects more often associated with the more severe and/or faster evolution of the disease in the CKCS, even by adding information for more advanced ACVIM classes and older subjects.”
The authors describe the study as cross-sectional – this is reasonable, but causal associations cannot be easily made with such a study design.
We thank you the reviewer for this comment and we agree that causal associations cannot be easily made using cross-sectional data. On the other hand, the authors have applied to veterinary medicine a method coming from the causal inference literature such as inverse probability weighting which attempt to infer causality weighting subjects in the study population. The weighting system aims to recreate a pseudo population where each subject is potentially receiver of all possible exposures (morphometrics) in this way emulating the randomization of the exposure. The randomization of the exposure is the key factor which allows to draw causal inference from clinical trials. Finally, even if the authors used a causal inference method to better understand the relationships investigated they have been cautious to use the word causality when interpreting the results from the analyses.
An alternative approach would be to take a population of CKCS with varying degrees of disease severity (from mild to severe) based on traditional measures of severity – LA volume, LV volume etc – and compare the morphometric characteristics in age matched and sex-matched dogs. In other words, do four 5- year old male dogs that have zero, mild, moderate and severe disease differ? With a large enough population that is age-andsex-matched, the authors might find some traits that change with disease severity. Finally, to show that any traits are associated with severity or “malignancy” the authors would need to perform a longitudinal study demonstrating that those dogs with shorter bodies, shorter nose and shorter heads developed CHF sooner than agematched counterparts with longer noses, bodies and heads.
Thank you for these suggestions. This is a very interesting approach, and we will surely consider it for the follow up, for which we have already started to collect data. Thank you.
Minor comments
Table 2 – many of the data are missing, at least based on the Table title. No mitral valve values were provided to this reviewer. I could only see morphometric data.
Thank you for this comment, this was a mistake during the text formatting. Thank you very much. We have added these data.
We feel that the quality of our manuscript has improved following the Reviewers’ comments and suggestions Thank you very much for you work and for bringing out some critical points.
Best regards

Reviewer 3 Report
The Authors present an interesting and original study about the influence of morphotype on development and progression of myxomatous mitral valve disease in Cavalier King Charles Spaniel, in class B1 according to with guidelines of ACVIM. Although the study has well-structured and results are very interesting for veterinary cardiologists, further improvements are necessary to make the manuscript eligible for publication.
Dogs presented a different age to the time of diagnosis, I suggest considering that as a limitation of the study, and it should be reconsidered.
In section M&M the authors describe inclusion criteria, considering that the number of dogs evaluated in B1 class is fifty-two. In subsection “Exclusion criteria” the authors confuse the lecturer, adding the number of dogs excluded (i.e. dogs in class B2). Please, delete the subsection or clarify. At line 114 add the name of the operator performing the echocardiographic examination. At line 426 is present a type (3).
An important revision of the English language is necessary; often sentences are very long, and the use of articles is not always correct.
I suggest using in the discussion the third person, and not the first plural.
In my opinion the paper is not adequate for publication, it needs major revisions.
Author Response
The Authors present an interesting and original study about the influence of morphotype on development and progression of myxomatous mitral valve disease in Cavalier King Charles Spaniel, in class B1 according to with guidelines of ACVIM. Although the study has well-structured and results are very interesting for veterinary cardiologists, further improvements are necessary to make the manuscript eligible for publication.
The authors thank the Reviewer for the review of our study. We are very pleased about the appreciation of the structure of the work and for considering the results interesting for cardiologists. We have carefully considered all Reviewer’ comments and have tried to address them whenever we felt this was appropriate.
Dogs presented a different age to the time of diagnosis, I suggest considering that as a limitation of the study, and it should be reconsidered.
Thank you for this comment. As you suggested we have added this in the discussion section: “It is also necessary to point out that the included subjects had a different age at the time of diagnosis.”
We agree with the reviewer that age is an important confounding factor in this study and the analyses have been performed adjusting for the confounding factors “age, sex, weight, and coat” which were used as regressors in the propensity score model. The propensity score model has been furtherly used to weight subjects in the study population on the base of such confounding factors making them more comparable and avoiding the confounding bias deriving by differences in age, sex, weight and coat of the dogs included in the analysis. With this method the differences in age at diagnosis should not constitute any longer a source of bias.
In section M&M the authors describe inclusion criteria, considering that the number of dogs evaluated in B1 class is fifty-two. In subsection “Exclusion criteria” the authors confuse the lecturer, adding the number of dogs excluded (i.e. dogs in class B2). Please, delete the subsection or clarify.
Thank you for this comment. We decided to specify the number of subjects of the initial population at the beginning of the inclusion criteria section: “The initial population of CKCS was composed by 72 dogs.”
We hope to be more precise adding this information.
We would like also to specify (as reported in exclusion criteria section) that subjects with diagnosed arrhythmias (n.4), as well as subjects with hypertension (n.3) or metabolic diseases (n.5) were not included in the study and belonged to ACVIM C class.
At line 114 add the name of the operator performing the echocardiographic examination.
Thank you, we have added this information.
At line 426 is present a type (3).
Thank you, we have corrected it. Thank you very much.
An important revision of the English language is necessary; often sentences are very long, and the use of articles is not always correct.
Thank you for this comment, we have required a certified English revision. Thank you.
I suggest using in the discussion the third person, and not the first plural.
Thank you very much for this comment, we have changed it and we have now used the third person.
In my opinion the paper is not adequate for publication, it needs major revisions.
Thank you for your thoroughly review of our study. We hope to have addressed al your requirements. Thank you.
We feel that the quality of our manuscript has improved following the Reviewers’ comments and suggestions Thank you very much for you work and for bringing out some critical points.
Best regards

Round 2
Reviewer 3 Report
Dear Authors,
Requested changes were performed and made revisions have considerably improved your manuscript.
It is my opinion that the paper is adequate for publication.
Yours sincerely
Author Response
The authors thank the Academic Editor for the thoroughly and rapid review of our manuscript.
We have carefully considered all comments and have tried to address them. We are very grateful of the Editor and Reviewers’ constructive comments and for the positive decision of Reviewer 3 after our first revision. We are very pleased.
Best regards